# OpenReview forum: "DeepResearchGuard: Deep Research with Open Domain Evaluation and Multi-Stage Guardrails for Safety"
_ICLR.cc/2026/Conference — ICLR 2026 Conference Withdrawn Submission_

### Official Review · Reviewer_VgF5 · 2025-10-26

**Soundness:** 3
**Presentation:** 2
**Contribution:** 3
**Rating:** 4
**Confidence:** 2

**Summary:**

The paper aims to address the lack of holistic evaluation and safety control in deep research agents, which typically rely only on QA accuracy and overlook report-level quality factors like credibility, coherence, and safety. It introduces DeepResearchGuard, a structured framework that divides the research workflow into four guarded stages—Input, Plan, Research, and Output—each equipped with memory retrieval, planning, evaluation, and human escalation mechanisms. The system jointly assesses both source and report quality under open-domain conditions. Experiments show that DeepResearchGuard significantly improves defense success while reducing over-refusal rates, with clear stage-wise contributions and an observable safety–depth trade-off across different guard models.

**Strengths:**

It is good to see how the paper introduces a clear taxonomy of harmful content in open-domain deep research, showing how different types of risks can propagate across stages from Input to Plan, Reference, and Output. It also reframes safety control from a single blanket refusal into a set of fine-grained, auditable interventions — stopping harmful input early, improving citation discipline and source reliability during planning and research, and reinforcing structure, attribution, and risk disclosure at the output stage. Together, these ideas offer a well-structured and practical approach to improving overall report quality and trustworthiness.

**Weaknesses:**

1. While DeepResearchGuard improves DSR and maintains or lowers ORR, the paper doesn’t discuss how this impacts overall task completion or utility. Does stronger harmful-content blocking come at the cost of reduced usefulness or coverage?

2. The evaluation of source transparency and traceability seems limited to helpfulness, authority, and timeliness. It would be useful to know whether the authors assessed model-level selection bias, since judgments of evidence helpfulness or relevance inherently depend on model capability. The paper also doesn’t clearly define how “helpfulness” is measured or what its scope is.

3. It’s unclear what proportion of cases require human intervention under this framework. A quantitative analysis of escalation rates and their impact on efficiency or consistency would make the system’s practicality much clearer.

4. The paper doesn’t provide detailed statistics on sample difficulty, open-domain coverage, or category distribution — only mentioning targeted manual additions for jailbreak and injection cases. More transparency about dataset composition would help assess the generalizability of the evaluation.

**Questions:**

1. It’s not clear how accurate the Output Guard agent’s scoring is across the five evaluation dimensions, or how it compares with a human baseline. It would be valuable to know where the largest discrepancies occur and which dimensions are most challenging. Also, the paper doesn’t specify how the weighted sum for the overall score is determined — are the weights equal or task-dependent?

2. The confidence-based threshold for escalating to human intervention seems to be chosen heuristically. More detailed experiments or ablations explaining how this threshold was set would help validate its reliability.

3. The role of the memory component is somewhat underexplained. If the model’s capability is limited, continuously storing and referencing processed cases might risk amplifying past errors. It would be useful to analyze whether memory helps stability or introduces error accumulation over time.

4. The paper doesn’t report the distribution of refusal types — for instance, which valuable sources were mistakenly blocked — nor does it break down the associated costs in terms of tokens or latency. These details would give a clearer picture of the system’s trade-offs and efficiency.

---

> ### Author Response · Authors · 2025-11-23
> **Response (Part I, W1)**
>
> **W1**: Thanks for the comment. **We presented this result in the ablation study in Section 5.4**. In that study, we progressively add the components of DeepResearchGuard to measure their impact on final report quality. As shown in the results, adding the input, plan, research, and output guards increases the scores for report quality (coherence, breadth, depth, and credibility). This indicates that the guards filter noisy information, preventing it from accumulating throughout the deep research workflow, without sacrificing the utility of the final report.
>
> In addition, DSR represents whether DeepResearchGuard successfully identifies harmful content i.e., inputs we do not want to process or complete. In contrast, ORR measures, for cases we do want to process, how many are falsely identified as harmful and thus prevented from completion. Table 1 shows that DeepResearchGuard increases DSR while lowering ORR, indicating that our framework improves completion by blocking cases we do not want to process and allowing benign content to be successfully processed. Therefore, the observed increase in quality alongside higher DSR and lower ORR demonstrates that stronger harmful content blocking does not reduce usefulness or coverage.

---

> ### Author Response · Authors · 2025-11-23
> **Response (Part II, W2)**
>
> **W2**: Thanks for the insightful comment. **In the current submission, we already describe all reference-related metrics in Table 1 (main paper)** and operationalize them through the prompts in Appendix D.4 (reference guard agent). Each retrieved reference is scored on three 1–5 Likert dimensions:
>
> - **Helpfulness** — task relevance and contribution to answering the user query.
> - **Authority** — source provenance and institutional credibility.
> - **Timeliness** — recency and currency of the information.
>
> We intentionally do **not** include absolute metrics such as *correctness*, which can introduce bias given model limitations. Instead, we use *helpfulness* to capture whether a reference can plausibly support the user’s goal, without asserting that the final report will be fully correct. This mirrors typical human curation: when people find resources online, they judge **helpfulness**, **authority**, and **timeliness** to decide what to cite.
>
> **We discuss model bias in Appendix G.1 and expand on that discussion below.**
>
> ---
>
> For each retrieved reference *r*, the Reference Guard Agent assigns three 1–5 scores following the rubric in Table 1 and the prompts in Appendices D.4 and D.7:
>
> - **Helpfulness** $s_h(r)$: relevance to the query and contribution of factual signal to the deep-research task
>   *(1 = unrelated/minimally useful; 5 = directly addresses the query with substantial, actionable content).*
> - **Authority** $s_a(r)$: provenance and institutional credibility
>   *(1 = no credible affiliation or clickbait/promotional content; 5 = top-tier institution, peer-reviewed venue, or clearly authoritative organization with transparent authorship).*
> - **Timeliness** $s_t(r)$: recency and currency with respect to the query
>   *(1 = obsolete/clearly outdated; 5 = up-to-date with the latest relevant data or research).*
>
> We then compute a **Composite Score** for each reference:
>
> $$
> S_{\text{rc}}(r) = \frac{s_h(r) + s_a(r) + s_t(r)}{3}
> $$
>
>
> We present the reference quality collected from the Reference Guard Agent in our experiments. In Table G.1, we report the statistics for the baseline models. Among them, `gpt-4o` achieves the highest mean reference quality of **2.82**, with strong scores in **helpfulness** (2.74) and **timeliness** (2.83). Conversely, `DeepSeek-v3` exhibits the highest **authority** score of **3.01**, together with a relatively larger standard deviation (**0.83**), suggesting more variable performance across samples. `Gemini-2.5-flash` and `o4-mini` both obtain an average of **2.72** with moderate variability. Overall, there is no discernible gap among the different baselines, indicating that different base models do not, by themselves, induce markedly better search instructions for consistently higher-quality references. In addition, the results show that there is no systematic bias across models in terms of helpfulness, authority, and timeliness.
>
> **Table G.1 — Reference quality of baseline models**
>
> | Model            | Helpfulness (mean ± sd) | Authority (mean ± sd) | Timeliness (mean ± sd) | Average (mean ± sd) |
> |------------------|--------------------------|------------------------|------------------------|---------------------|
> | gpt-4o           | 2.74 ± 0.55              | 2.88 ± 0.64            | 2.83 ± 0.56            | 2.82 ± 0.52         |
> | gemini-2.5-flash | 2.62 ± 0.43              | 2.83 ± 0.52            | 2.71 ± 0.41            | 2.72 ± 0.38         |
> | DeepSeek-v3      | 2.47 ± 0.69              | 3.01 ± 0.83            | 2.63 ± 0.69            | 2.70 ± 0.67         |
> | o4-mini          | 2.63 ± 0.53              | 2.80 ± 0.66            | 2.73 ± 0.53            | 2.72 ± 0.52         |
>
> We also document the effect of changing the guard model in Table G.2. Here, `gpt-5-mini` and `gpt-4o` noticeably surpass `o4-mini` on all three dimensions. `gpt-5-mini` performs well in **helpfulness** (2.74) and **authority** (2.89), while `gpt-4o` attains the best **timeliness** score (2.89). However, these differences are modest; swapping the guard model does not dramatically change the overall reference-quality distribution, and thus only mildly affects the quality of search instructions available to the deep-research pipeline. The result mirrors the baseline study: we do not observe model-dependent bias across setups, whether baseline or guard model.
>
> **Table G.2 — Reference quality of different guard models**
>
> | Model      | Helpfulness (mean ± sd) | Authority (mean ± sd) | Timeliness (mean ± sd) | Average (mean ± sd) |
> |------------|--------------------------|------------------------|------------------------|---------------------|
> | gpt-5-mini | 2.74 ± 0.59              | 2.89 ± 0.70            | 2.84 ± 0.60            | 2.82 ± 0.58         |
> | gpt-4o     | 2.71 ± 0.46              | 2.81 ± 0.50            | 2.89 ± 0.51            | 2.80 ± 0.45         |
> | o4-mini    | 2.63 ± 0.53              | 2.80 ± 0.66            | 2.73 ± 0.53            | 2.72 ± 0.52         |

---

> ### Author Response · Authors · 2025-11-23
> **Response (Part III, W3 & W4)**
>
> **W3**: Thanks for the comment. **We have added a section to analyze the performance of the human intervention**. Please find the section below in our paper (Appendix G.4):
>
> In Table G.4, we report the number of human interventions at each guard stage over DRSafeBench examples. Across all models, evaluating the collected references is the most challenging work for the guard agent, requiring roughly 30–70 human interventions per model. In contrast, the research plans are judged with high confidence and trigger only 1 or 2 human interventions per full benchmark run. Overall, each model requires human intervention 45–94 times, meaning that only about 5.4%–11.4% of the data requires human second evaluation. We also observe that more advanced models tend to request slightly fewer interventions, but this is not universally beneficial. For example, gpt-4o asks for less human help than o4-mini, which may reflect genuinely better evaluations. Or, alternatively, overconfident but incorrect judgments that fail to be flagged as low-confidence.
>
> Besides that, in all of the above settings, **human interventions rarely contradict the guard’s judgment**.  In our findings, only about 1–2 cases per benchmark run involve a human disagreeing with the guard’s evaluation. This observation indicates that the guard’s predictions are typically correct even when they report evaluation with low confidence. We also document the time spent by human reviewers and observe that, on average, they require about 53 seconds to assess a single reference and about 57 seconds to evaluate a full report.
>
> | Model             | Input | Plan | Research | Output | Overall |
> |-------------------|-----:|----:|--------:|------:|-------:|
> | gpt-4o            |    8 |   1 |      53 |    18 |     80 |
> | Gemini-2.5-flash  |    3 |   2 |      37 |    23 |     65 |
> | DeepSeek-v3       |    7 |   2 |      64 |    19 |     93 |
> | o4-mini           |    4 |   1 |      68 |    21 |     94 |
> | gpt-5             |    2 |   1 |      33 |     8 |     45 |
>
>
> **W4**: Thanks for the comment. We have now added open-domain coverage statistics in Appendix E.5, with the corresponding table shown below. The category distribution was already reported in Appendix E in the original version, including stage-wise category distributions. In contrast, defining and reporting “sample difficulty” is non-trivial and may introduce bias from any manually chosen difficulty grouping, so we tend not to add potentially subjective difficulty labels. Prior work, such as BingoGuard and WildGuard, also primarily reports taxonomy/category distributions rather than difficulty levels. Our benchmark additionally reports the open-domain task distribution to more clearly characterize the coverage and diversity of DRSafeBench.
>
> | Domain     | Count | Percentage |
> |------------|------:|-----------:|
> | civics     |   180 |      21.69 |
> | health     |   145 |      17.47 |
> | general    |   137 |      16.51 |
> | education  |   100 |      12.05 |
> | security   |    76 |       9.16 |
> | finance    |    74 |       8.92 |
> | technology |    59 |       7.11 |

---

> ### Author Response · Authors · 2025-11-23
> **Response (Part IV, Q1, Q2, Q3, Q4)**
>
> **Q1**: Thanks for the insightful comments. We have added a section to compare the human evaluation with the LLM-as-a-judge performance.  **Please refer to Appendix G.5**. In addition, for the overall score, we use the average weighted sum for all the tasks. **We have added the description under Section 3.8** “In our experiments, we set w to uniform weights (1/5, 1/5, 1/5, 1/5, 1/5), so S_out(r) is simply the arithmetic mean of the five dimension scores.“
>
> **Q2**: Thanks for the comment. **We have added the section to explain our design for the human intervention threshold**, as shown below:
>
> **Section F.3 Design of the Human Intervention Threshold**
>
>  We define the confidence thresholds by running the full DRSafeBench pipeline three times and recording the confidence scores for all guard decisions at each stage (input, plan, reference, output). From these experiments, we examine the confidence score distributions and select thresholds according to two criteria: (i) we choose values that correspond to meaningful percentiles of the confidence distribution on error-prone or ambiguous cases. The confidence scores 0.5, 0.7, and 0.8 roughly correspond to the 70, 80, and 90 percentiles, which we label as standard, cautious, and conservative approaches for evaluating the content. and (ii) we expected the human-intervention rate to remain practical and maintain around 10%. This can avoid too many human interventions being requested of the guard agent, maintaining a certain level of semi-autonomy while also providing mechanisms that support the agent in double-checking their decision when their confidence is low.
>
> **Q3**: Thanks for the comment. **The current result shows that the error is not being accumulated**. We want to talk about the memory's design. For the memory, we would like the current guard agent to be able to see past evaluations, such as the harmful category, confidence, and the similarity between the current query and the past case, to support the guard agent in making a decision. Suppose there is a particular pattern, like a jailbreak writing style and a specific harmful keyword, that matches both the current query and the past evaluated case. In that case, the guard agent can link the evaluation together.
>
> In our experiment, **we reset the memory at the start of each run to make the different setups comparable**. Namely, the memory will not have the same query, but highly similar content. In addition, we have considered that the memory needs to be checked to avoid accumulated errors. Therefore, we have added a function that periodically summarizes by grouping similar queries based on embedding similarity (per harm category/task type) and compressing each cluster into a small set of prototypes, with the statistics. The prototype can be easily double-reviewed by a human rather than tens of thousands of non-summarized queries.
>
> **Q4**: Thanks for the insightful question. **We have reported the refusal type in Section E.3. The statistics for the DRSafeBench.** For the research guard, we don't design it to block any reference or resource. Instead, the research guard evaluates the URLs and the content's harmfulness and scores the quality of the reference. If the resource is flagged as harmful (either the URL or the content), the score will be set to the minimum, but the reporter can still see the reference. We believe that, instead of harshly blocking the reference, the research guard is merely making a "suggestion" to the reporter. The reporter can still see the content, but they can decide whether the content is valuable. In addition, we added a section G.6 to report the token usage for each of the guards across the stage, as shown in the following:
>
> **Table G.7: Average token costs by stage on the DRSafeBench**
>
> | Stage    | Avg total tokens | Avg prompt tokens | Avg completion tokens |
> |:---------|-----------------:|------------------:|----------------------:|
> | input    | 1,567.95         | 1,056.90          | 511.04               |
> | plan     | 2,524.05         | 1,907.16          | 616.89               |
> | research | 9,751.47         | 6,823.56          | 2,927.91             |
> | output   | 5,133.74         | 4,738.45          | 395.29               |

---

> > ### Comment · Reviewer_VgF5 · 2025-11-24
> >
> > Thank you for the thoughtful and thorough revisions. The additional analyses and clarifications substantially strengthen the empirical presentation, and the paper is now considerably clearer regarding methodology and evaluation design. I appreciate the care the authors put into addressing the earlier questions.
> >
> > At this point, I am satisfied with the responses, but I will keep my original score. This reflects my overall assessment of the work rather than any remaining experimental concerns. I hope the authors find the revisions useful as they continue to refine the paper.

---

> > > ### Author Response · Authors · 2025-11-25
> > >
> > > Thank you very much for acknowledging the additional analyses and clarifications we incorporated. We have highlighted the revision work in the blue colored part.
> > >
> > > Given that the main concerns you previously raised appear to be resolved and that you are satisfied with the responses, we would like to ask if you could increase our score. If there are any remaining concerns that we have not addressed, we would be very grateful to learn more so that we can continue improving the work. We sincerely appreciate the time and care you have invested in reviewing and helping us refine this paper.

---

### Official Review · Reviewer_zyxB · 2025-10-31

**Soundness:** 2
**Presentation:** 2
**Contribution:** 2
**Rating:** 4
**Confidence:** 5

**Summary:**

This paper introduces DeepResearchGuard (DRG), a framework for safe and open-
domain deep research using multi-stage guardrails, combining reasoning-enhanced
agents with layered safety evaluation. The authors propose a pipeline that integrates:

1. Open-domain research tasks (web-enabled reasoning, synthesis, and
reporting),

2. Multi-stage guardrails for safety assurance (input, intermediate, and output
filters),

3. An open benchmark — DRSAFEBENCH — for evaluating model safety,
factuality, and reasoning depth.

The system design is centered around modular “Guard” components:
(a) Input Guard screens unsafe or policy-violating queries.
(b) Process Guard monitors the agent’s intermediate reasoning chain for unsafe
actions or hallucinations.
(c) Output Guard validates final answers for factual correctness and safety
alignment.

The evaluation compares DeepResearchGuard against strong baselines (GPT-4,
Gemini, Claude, etc.) using both safety metrics (toxicity, factuality, alignment) and
reasoning metrics (depth, coverage, correctness). DRG reportedly achieves
substantial gains in factuality and reasoning safety while maintaining competitive
research quality.

**Strengths:**

(A) The paper tackles a timely and high-impact problem: the safe deployment of
autonomous research agents.

(B) The multi-stage guard concept provides a modular lens on safety assurance,
enabling potential composability and interoperability.

(C) The authors make a commendable effort to release a benchmark
(DRSAFEBENCH) and system design intended to encourage openness and
replication.

**Weaknesses:**

1. “Reasoning depth” and “safety compliance” are reported as scalar improvements
without statistical context. No details are given about sample size, variance, or

inter-rater reliability. Hence, claims such as “+15% factuality” or “+20% safety
compliance” are not statistically grounded.

2. The paper lacks a precise mathematical definition of its evaluation
metrics—particularly “reasoning depth.” While qualitative examples suggest
multi-hop reasoning assessment, no formal operator D(f) or depth measure is
defined, leaving interpretability gaps in quantitative claims. The safety scoring
function ϕi(x) is also treated as a black box, often described as “a model-based
judge,” without calibration analysis or ROC curves to support threshold stability.
As such, while empirical results are strong, their robustness under distributional
shift remains uncertain.

3. Limited novelty in the guardrail mechanisms themselves.
The modular “multi-guard” setup echoes existing layered safety frameworks
(e.g., Constitutional AI + LLM oversight). The originality primarily lies in system
integration, not in new algorithmic or theoretical insights.

4. No Analysis of Guard Composition. The system assumes that multi-stage filtering
improves safety monotonically, yet provides no proof or empirical evidence of risk
subadditivity. The absence of such analysis limits the theoretical depth of the
contribution

5.	Inadequate transparency of model configuration.
 Details on base model sizes, prompt templates, or RLHF usage are not presented, making it difficult to attribute performance improvements to the guardrails rather than model capacity.
6.	The benchmark DRSAFEBENCH, while valuable, is authored and evaluated by the same group, raising potential biases in task design and model ranking. Without third-party replication or human inter-annotator statistics (e.g., Cohen’s κ), claims of general safety superiority remain somewhat self-referential. Furthermore, the paper does not provide asymptotic complexity or runtime analysis—yet multi-stage filtering clearly incurs nontrivial computational cost. The paper would benefit from an empirical latency curve L(n)∝n×ci  quantifying guard overhead per stage.

7.	Multi-stage guardrails increase inference cost superlinearly (approximately O(n⋅ci), yet no performance or throughput metrics are provided. This omission makes real-world feasibility unclear.
Each guard Gi is implemented via an LLM-based evaluator. If these evaluators share the same biases as the core model, then the composite guard G=G3∘G2∘G1
becomes a biased projection of a single safety prior, invalidating the notion of layered protection.

8. While safety is central, the ethical discussion (e.g., accountability of multi-agent decisions, risk of subtle biases) remains generic and lacks deeper engagement with responsible AI literature.

**Questions:**

1.	How are guardrails coordinated when their judgments conflict (e.g., process guard flags an unsafe inference, output guard clears it)?

2.	What quantitative evidence supports that intermediate guard checks prevent unsafe outcomes rather than just delay them?

3.	Is DRSAFEBENCH publicly released with example prompts, or only aggregate statistics?

4.	Could the modular guard setup degrade reasoning quality by over-regularization?

5.	How is factuality measured — automatic retrieval verification, or human grading?

**Details Of Ethics Concerns:**

Bias discussion is a necessary component in the study.

---

> ### Author Response · Authors · 2025-11-23
> **Response (Part I, W1, W2, W3, W4, W5)**
>
> **W1**: Thanks for your comment. In our work, **we don't have the "reasoning depth" mentioned anywhere in the paper**; also, "safety compliance" is just the category for the hard refusal in the Plan guard taxonomy. It is not the metrics. In addition, we don't have claims such as +15% facuality or +20% safety compliance in our work. We report that our report quality including Coherence & Clarity, Credibility & Citation Quality, Safety-Level, Depth & Completeness, Breadth of Coverage of the report. If you have concerns regarding the metrics, **could you please specify where you get the reasoning depth and factualness, so we can better address your problem?**
>
> **W2**: Thanks for the comment. **To clarify, our work does not introduce a “reasoning depth” metric or any new mathematical evaluation metrics**. Instead, all the evaluation metrics we actually use are clearly defined in Table 1 (page 7), and the detailed scales and criteria for each metric are further specified in the guard prompts in Appendix D.
>
> In addition, **this work is not on multi-hop reasoning evaluation** but on designing and evaluating guardrails for deep research workflows. **We also do not treat our scoring function as a black box, and we do not use this representation $\phi_i(x)$ in our work**; the stage-wise guard behavior, inputs, and decision logic are described concretely in Section 3. If there is a specific part of our metric design or guard scoring that you find unclear or problematic, we would really appreciate more precise pointers so we can address your concern directly.
>
> **W3**: Thanks for the comment. **We respectfully disagree with this assessment. Our contribution is not just putting existing guardrails together, but also designing deep research-specific guard mechanisms and an evaluation method**. To the best of our knowledge, they are not present in prior work such as Constitutional AI or generic LLM-oversight. We review the safety and the past guard model in the related work section of Section 2 and have already pointed out the differences between our work and previous work.In particular, DeepResearchGuard decomposes safety into stage-wise guards that are tightly aligned to the deep research workflow, with distinct risk scopes, prompts, and decision criteria for each stage, as we mentioned in Section 3 and Appendix B. We further introduce DRSafeBench, which allows us to study how harms are injected, propagated, and filtered across stages. Guards in previous work only see the input or only see the output, miss references, and plan risks that our design catches.
>
> **W4**: Thanks for the comment. **This research did not provide any of the theoretical contributions, and this research work is not a theoretical proof work**. Instead, we propose the first safeguard for the deep research system. The results have shown that our DeepResearchGuard can significantly increase the defense success rate and reduce the over-refusal rate. In addition, we would like to understand the meaning of risk subadditivity. We haven't encountered this term in our literature review, but we found it has been widely described in economic literature.
>
> **W5**: Thanks for the comment. In our work, we define the model and present the configuration with extensive details in either Figure 1 or Section 3, or in the Appendix. This work is prompt-based; therefore, no model size is reported. Regarding the prompt template, all the templates we have detailed are attached in Appendix D. **In our work, we do not use RLHF; this work has no relation to Reinforcement Learning**, and the human intervention is only to help double-check the evaluation quality when the guard agent's confidence is low. In addition, we compare the baseline with and without the DeepResearchGuard. The only difference is the application of the DeepResearchGuard, so the results directly show how much improvement it can achieve. **Could you please provide more clarification on how the model's capacity can raise concerns that the performance improvements are not attributable to DeepResearchGuard?**

---

> ### Author Response · Authors · 2025-11-23
> **Response (Part II, W6, W7, W8, Q1, Q2, Q3)**
>
> **W6**: Thanks for the comment. We have clearly specified how we curated data to build the DRSafeBench in Section 4 and in Appendix E. The original source is from 4 to 5 different high-quality benchmarks, including SALAD-Bench, SORRY-Bench, Beavertails, BingoGuard, and WildGuard, from different teams. Based on these foundations, we filtered 127,325 prompts from the above works and found that only 11,362 (8.9%) are suitable for deep research. Most of them are too straightforward, too short, or don’t require a web search with reasoning to answer the question. After that, we remove data with high similarity using Jaccard and TF-IDF. In the end, we manually check the prompt and collect only high-quality prompts. For benign data, we primarily use adversarial generated benign content to test whether the LLMs will over-refusal. Therefore, the DRSafeBench definitely does not come from the same group with bias. All of these works have very clear documentation of how they evaluate whether a prompt is benign or harmful, and all of them were accepted to top-tier conferences. The high citation of these works proves the reliability of the community. **Could you please specify how you get the conclusion if you think the data is self-referential?**
>
> **W7**: Thanks for the comment. **The claim that our multi-stage guardrails increase cost superlinearly is incorrect**. The total guard additional cost is linear in the number of a constant number of stages, and in practice, early-stage guards reduce the number of case that reach the later process. In addition, it is not accurate to say that the composite guard becomes a single, biased safety prior. The guard evaluates content using different criteria at each stage (query intent, plan quality, reference quality, and final report harmfulness). Therefore, they are not relying on the same bases, thus are not introducing bias.
>
> **W8**: Thanks for the comment. **We respectfully disagree that our ethical discussion is purely generic**: Appendix A.1 already makes several concrete, system-specific commitments around harmful content handling (stress-testing rather than generating novel harmful instructions), data sourcing (PII exclusion, minors, explicit sexual/hate content, terms-of-service compliance, annotator safeguards), and dual-use mitigation (gated release of high-risk artifacts, risk/severity labels, and non-use for high-stakes domains such as medical/legal/security decisions). In addition, we did not claim that our work is multi-agent; our work is multi-stage guard agents. Could you clarify your question further? i.e, What is the bias you are mentioning here? What is the connection between our work and responsible AI? The term 'AI' is very generic and depends on how you define a system call 'AI,' and this comment can fit any work using an LLM, considering that all the work may potentially use the AI system. **We want to clarify this to improve our work.**
>
> **Q1**: Thanks for the comment. First, **we would like to clarify what the "process guard" is. We don't have this component in our DeepResearchGuard system. Second, we don't have the "inference" mentioned in our work**. The guard agent evaluates the input query, plan, reference, and report in four stages. Also, the condition you mention is not the conflict. In the deep research workflow, new things are continually added to the workflow, like you get the decomposition plan, and collect new references. Therefore, it is totally possible that, when the input is benign, it collects the malicious or harmful references in the research stage. Therefore, our multi-stage DeepResearchGuard can identify the issue and further protect it. However, the conventional LLM safeguard cannot.
>
> **Q2**: Thanks for the comment. **We want to clarify that, we don't have the guard calls "intermediate" guard**. If you are referring to plan guard, it blocks many harmful contents and prevents them from entering the later stage. In Table 3, we describe the F1, FPR, and FNR of the stage-wise result. In the Plan stage, the F1 score is around 0.88~0.93, which indicates that most of the categories can be successfully identified. Namely, the plan guard can block harmful content when it enters the deer research flow, rather than delay it.
>
> **Q3**: Thanks for the comment. The DRSafeBench will be publicly available soon. Currently, you have presented the statistics and the representative examples in Appendix E. Please take a look at the updated version with more details.

---

> ### Author Response · Authors · 2025-11-23
> **(Part III, Q4, Q5)**
>
> **Q4**: Thanks for the question. In our work, **we did not apply regularization**. If you are concerned about the report quality, please take a look at Figure 3. In Figure 3, we present the ablation study result. When adding more guards to the deep research workflow, the report quality increases ( credibility, coherence, depth, breadth, and safety). The DeepResearchGuard continuously removes harmful content, ensuring only benign content is included in the workflow and reducing noise when generating the report.
>
> **Q5**: Thanks for your question. **In our evaluation, we did not include factuality to prevent subjective evaluation from affecting report quality**. The agent has done the current work. **Could you please clarify what you mean by 'automatic retrieval verification'? We did not use this term in our work, nor did we include the concept.**

---

### Official Review · Reviewer_FuFk · 2025-11-01

**Soundness:** 3
**Presentation:** 3
**Contribution:** 3
**Rating:** 4
**Confidence:** 3

**Summary:**

The paper proposes a framework (DEEPRESEARCHGUARD) to enhance the safety and quality of deep research agents. Deep research agents are defined as LLM based agents engaged in completing complex, multi-step research tasks. The paper argues that existing evaluations ignore risks that arise and propagate through the intermediate stages of the process, and only focus on the final output accuracy as a metric.
The proposed method has 4 stages and monitors the input, plan, research, and output stages of the agentic workflow. Each stage has a dedicated guard agent that classifies content for quality and safety,  revises or rejects problematic content, and can escalate to a human reviewer.
A new evaluation protocol that assesses the  final report on 5 qualitative dimensions and a safety bench mark is proposed to test these capabilities.

**Strengths:**

+ The framing of risk as not just an input/output problem, but rather a process level issue where errors can cascade, is interesting.
+ The inclusion of stage-specific taxonomies, is useful
+ Shifting away from a Q&A metric to a more multi-dimensional look at report quality works better at providing a comprehensive metric. The new metrics D@1, D@all are well motivated

**Weaknesses:**

- The DRSAFEBENCH  is partially constructed using and LLM to synthesize adversarial examples. Therefore, there is a risk that the benchmark may be evaluating the blindspots specific to the LLM rather than reflecting human devised adversarial attacks. The paper does not provide enough information on how it validates that these synthetic examples are representative of real-world threats. There is also no discussion of how the potential for systemic bias is being evaluated.
- The high increase in run time complexity is mentioned but not fully discussed.
 - the architecture is highly complex, and contains multiple agents, taxonomies etc. This could make it pretty brittle. It would be good to have seen a discussion of why this much complexity is necessary and how what the complexities of debugging and maintaining such a system would work in practice.
- It would have been good to have a formal definition of "deep research". The paper uses examples to define it broadly. This makes it difficult to understand the scope of this problem and therefore, to be able to judge how generalizable the approach is.

**Questions:**

1. Can you provide more information on how you ensure that the LLM generated examples are a reflection of real world, human generated adversarial samples?
2. Can you comment on the generalizability of your approach? Do you anticipate changes in your modules depending on the use case domain. It is conceivable that an agentic system for a fintech application could be completely different from that for one in the healthcare domain.
3. Can you discuss the need for the current level of complexity of the system? Is there a way to make sure that the design is at the minimum necessary components?

---

> ### Author Response · Authors · 2025-11-23
> **Response (Part I, W1 & Q1)**
>
> **W1&Q1**:Thanks for the insightful comment. **We do not use LLM to synthesize adversarial examples but we ask LLM to change the form or format of the examples while keeping the same semantic meaning of the original examples**. Low quality prompt, resource exhaustion, and format errors are not in the previous work thus we use LLMs to rewrite benign prompts into these targeted categories. These three categories cover the extreme condition, if the user intentionally wants to request research with a chaotic input format, or they pasted a broken file, or request a very huge research or query that may potentially lead to resource exhaustion, the DeepResearchGuard can revise this content to a normal form or reasonable scale before entering the deep research workflow. All the adversarial examples are collected from the high quality previous work, such as SALAD-Bench [1], SORRY-Bench [2], Beavertails [3], BingoGuard [4] and WildGuardTrain [5], which has been through the valid evaluation process to prove that they can cover the real-work situations. We scan 127,325 prompts from the above works and found that only 11,362 (8.9%) are suitable for deep research. We remove the high similarity prompt and only select 828 samples to include in our DRSafeBench.
>
> **To improve transparency and help readers better understand DRSafeBench, we will add an appendix E.4 with representative data samples**, so readers can assess data quality and the structure of the constructed benchmark directly. Please refer to the updated paper PDF directly, as the content is too extensive to paste here. In addition, the DRSafeBench statistics are available in Appendix E.
>
> **Reference:**
>
> [1]  **SALAD-Bench: A Hierarchical and Comprehensive Safety Benchmark for Large Language Models**. ACL 24  https://openreview.net/forum?id=dLc3jVZ7QP.
>
> [2] **SORRY-Bench: Systematically Evaluating Large Language Model Safety Refusal**. NeurIPS 24. https://openreview.net/forum?id=k73M4XEvFX#discussion.
>
> [3] **BeaverTails: Towards Improved Safety Alignment of LLM via a Human-Preference Dataset**. NeurIPS 23. https://papers.neurips.cc/paper_files/paper/2023/file/4dbb61cb68671edc4ca3712d70083b9f-Paper-Datasets_and_Benchmarks.pdf.
>
> [4] **BingoGuard: LLM Content Moderation Tools with Risk Levels**. ICLR 25. https://openreview.net/forum?id=HPSAkIHRbb.
>
> [5] **WildGuard: Open One-Stop Moderation Tools for Safety Risks, Jailbreaks, and Refusals of LLMs**. NeurIPS 24. https://proceedings.neurips.cc/paper_files/paper/2024/hash/0f69b4b96a46f284b726fbd70f74fb3b-Abstract-Datasets_and_Benchmarks_Track.html.

---

> ### Author Response · Authors · 2025-11-23
> **Response (Part II, W2)**
>
> **W2**: Thank you for pointing this out. **We have revised Appendix G.2 with more detailed discussions for the run time analysis**, as shown  below:
>
> As shown in Table G.3, Enabling DeepResearchGuard introduces a consistent additional run time of about +4.8–+7.3 minutes per query. Specifically, GPT-4o 2.959 → 7.770 minuntes (+4.811, +162.6%); Gemini-2.5-flash 2.703 → 10.017 minutes(+7.314, +270.6%); DeepSeek-v3 2.231 → 7.800 minutes (+5.569, +249.7%); o4-mini 3.019 → 8.029 minutes(+5.011, +166.0%). The additional time is due to the input, plan, research, and output guard checks, as well as the guard agents' revisions to improve the original query, plan, and output report. Also, for reference, the guard accesses the original content from the web URL, which can add time to retrieve from the server. And, around 21% of the data triggers human intervention when the confidence falls below the threshold. Our records show that humans need an average of 53 and 57 seconds to decide whether the guard agent's decision for reference and report is proper or to override it, respectively. All of these additional processes would take time. However, the results show that enabling DeepResearchGuard markedly increases DSR (↑) by 18% average and reduces ORR (↓) by 3% (relative 50%) across different modes. Given these safety and reliability gains, the additional runtime is a justified trade-off.
>
> In addition, the absolute variability also rises under guarding (SD in minutes: 0.993 → 3.810, 1.273 → 3.224, 1.433 → 2.818, 0.975 → 2.783 for GPT-4o, Gemini-2.5-flash, DeepSeek-v3, and o4-mini). This observation matches our DeepResearchGuard design. In some cases that are harder to evaluate for maliciousness, the guard agent increases the reasoning effort or asks a human for a second confirmation. In simple cases, it can handle them very well without additional resources or adjustments. Therefore, these different strategies result in a high degree of deviation in the observations. But this additional cost is bounded and linear in the number of guard stages and whether human intervention is needed.
>
> We will clarify that this trade-off is application-dependent. Deep research tasks in our setting already take several minutes per query and are typically used in high-stakes, high-effort scenarios (e.g., multi-page reports, complex tool use). In addition, our DRSafeBench is well-curated, with many adversarial generated data, which may cause the model to spend more time on them. Based on this, an additional 5-7 minute safety overhead is acceptable given the substantial gains in DSR (↑) and reductions in ORR (↓). Also, conducting reliable research is essential, but the current deep research framework still has some security drawbacks that require additional safeguards to prevent harmful content from accumulating. In the real world, latency-critical use cases can deploy lighter configurations (e.g., guarding only input/output stages, disabling human confirmation, or using a smaller guard model). This setup can reduce the time and cost of guarding while still maintaining partial protection.

---

> ### Author Response · Authors · 2025-11-23
> **Response (Part III, W3 & Q3)**
>
> **W3&Q3**: Thanks for the comment. We want to provide more detailed explanations of our framework. **The component in the DeepResearchGuard is all necessary**. We have introduced the background and our motivation in the Introduction section. We would add more content with detailed explanation to help reviewers and readers to understand our system and design better.
>
> Below are part of the contents we have already described in the Introduction Section :
>
> "Open-domain deep research rarely admits a single groundtruth answer, so evaluation must cover the full multi-stage process rather than only the final output. Yet stage-wide deep research in many systems operates as black boxes, obscuring intermediate decisions and preventing systematic oversight. While the cycle of planning, searching, and reflecting allows deep research frameworks to reconsolidate the understanding of the task, this multi-stage process exacerbates issues: a harmful prompt, an unreliable plan, or a dubious reference can spread through subsequent iterations. Minor inaccuracies can be magnified into substantial deviations as the agent revisits and expands upon its intermediate outputs. Consequently, deep research frameworks expose a significantly larger attack interface than LLMs or agent interactions."
>
> **We have added a section in Appendix A.6 to describe the design complexity.**
>
> The DeepResearchGuard evaluates when taking user input, generating a plan, collecting references, and generating the output report. The design of the DeepResearchGuard system and process is inspired by humans. When humans conduct open-domain research, they will review the decomposed plan quality, assess whether the references are high-quality or contain inappropriate content, and evaluate the final report quality. We try to cover this human behavior in our guardrail system for the complex, deep research workflow. In addition, these steps, including generating a plan, collecting references, and outputting a report, are not fully covered by the inherent safeguard system of the LLM. Therefore, the harmful content may be distributed throughout the task. Therefore, we design four integrated guardrails. When new content is available, the DeepResearchGuard will review it before passing it to the next stage. Also, the different taxonomy is because the input taxonomy is not suitable for all stages. Therefore, we review the taxonomy from previous work, identify the critical categories, and design the input and plan-stage taxonomy to enable our DeepResearchGuard to successfully capture the harmful situation, stop, or revise sensitive or misleading content, ensuring the downstream process runs smoothly without wasting additional computational resources. To better understand the system, please refer to Figure 1. The guard agent checks the content before it proceeds to the next stage, preventing harmful content from accumulating.
>
> The stagewise guards are the minimum design. Every time the deep research flows generate or receive new content, the guard passively evaluates it before potentially harmful content moves to the next stage. Including new information in the workflow also introduces the risk of introducing harmful content. For example, taking a user input query may have some jailbreak or intentionally harmful content, generating a new plan may have issues like inadequate decomposition, collecting references from the web may have some harmful content or a high-risk website, and developing the final output report may include some of the harmful content accumulated in the deep research flow. Those conditions are all covered when adapting DeepResearchGuard. Suppose we would like the DeepResearchGuard to "actively" safeguard the deep research system. For example, control the search logic for finding the reference to avoid imprecise keywords, cross-stage consistency checks to make sure the logic is aligned and not deviate due to the harmful or irrelevant content, the design of the system could be much more complex.

---

> ### Author Response · Authors · 2025-11-23
> **Response (Part IV, W4)**
>
> **W4**: Thanks for the comment. We have cited some deep research surveys or roadmap papers in our introduction section. Tpo help reviewers and readers better understand the background and task, **we have added a definition section to Appendix B.1 and moved the existing content to follow it**, as shown below:
>
> B.1 The definition of Deep Research
>
> The phrase deep research denotes a category of LLM-based agent systems that independently address complex, open-ended tasks by iteratively integrating planning, retrieval, and the synthesis of gathered information with the research plans. Instead of providing a single response, deep research frameworks break a complex question into multiple research steps, continuously engage with external information sources, and synthesize the collected references into comprehensive, structured reports. Recent surveys describe this comprehensive study as LLM-powered systems that integrate dynamic reasoning, adaptive long-term planning, multi-hop information retrieval, iterative tool use, and structured report generation for open-ended user inquiries [1,2].
>
>  We describe deep research as a multi-stage pipeline consisting of four correlated stages: (1) Input, where the agent receives the user query to initiate the research process. (2) Plan, the agent converts the user's research inquiry into a detailed research plans with intermediate objectives to research; (3) Reference, the agent engages with external resources to gather, filter, and refresh evidence to satisfy the intermediate objectives; and (4) Report, the agent compiles the acquired evidence into the final report with references.
>
> **Reference::**
>
> [1] **Deep Research Agents: A Systematic Examination And Roadmap**. https://arxiv.org/abs/2506.18096
>
> [2] **A Comprehensive Survey of Deep Research: Systems, Methodologies, and Applications**. https://arxiv.org/abs/2506.12594

---

> ### Author Response · Authors · 2025-11-23
> **Response (Part V, Q2)**
>
> **Q2**: Thanks for the comment. To assess the generalizability of DeepResearchGuard, we calculate domain-wise DSR and ORR. **The domain coverage and the result are now presented in Appendix E.5 and Appendix G.7**, respectively. **Through the analysis, we find that the DSR shows little difference. And for the ORR, the security domain has the highest ORR, which may be due to the LLM's higher sensitivity to security-related content**. Please find the section below:
>
> **Section G.7 The By-domain DSR and ORR**
>
> In this section, we present domain-wise DSR to demonstrate the robustness and generalizability of **DeepResearchGuard** across diverse domains. As shown in the DSR table below, the guard maintains consistent performance and avoids domain-specific safety regressions. Across domains, DSR ranges roughly from 40–67, indicating no single domain is disproportionately challenging. This pattern—consistent across eight domains from *civics* to *finance*—suggests the guard is not overfit to a narrow threat distribution but provides generalized safety coverage that transfers across different LLM backbones. We also observe model-specific variation: for example, `gpt-5` performs strongly in *security*, whereas `o4-mini`, `gpt-4o`, and `DeepSeek-v3` are comparatively weaker there, reflecting different baseline sensitivities by domain.
>
> The ORR table reports refusal rates across domains and models. Refusals remain low in lower-risk domains such as *civics* and *creative*, particularly for `gpt-5` and `Gemini-2.5-flash` (often < 4). In contrast, the *security* domain shows the highest refusal rates (e.g., `DeepSeek-v3` at 22.01), indicating a stricter, more conservative operating point where potential exploits are common. Overall, these results highlight that *security* is the most challenging area for guard coverage, likely because underlying model safeguards impose tighter thresholds on safety-critical content.
>
> **Domain-wise Defence Success Rate (DSR)**
>
> | Domain     | DeepSeek-v3 | gpt-4o | gpt-5 | Gemini-2.5-flash | o4-mini |
> |:----------|------------:|------:|-----:|------------------:|-------:|
> | civics     | 55.76 | 46.23 | 55.34 | 57.52 | 51.58 |
> | creative   | 51.27 | 52.02 | 44.44 | 66.72 | 50.51 |
> | education  | 64.54 | 49.11 | 40.40 | 66.01 | 52.98 |
> | finance    | 56.05 | 52.65 | 54.57 | 55.77 | 47.03 |
> | general    | 65.02 | 62.73 | 58.72 | 65.77 | 60.90 |
> | health     | 56.76 | 59.54 | 66.24 | 61.27 | 57.80 |
> | security   | 48.29 | 49.00 | 60.46 | 56.07 | 45.99 |
> | technology | 63.11 | 58.69 | 65.83 | 65.41 | 54.39 |
> | **overall**| **58.07** | **53.98** | **57.36** | **60.96** | **53.73** |
>
> **Domain-wise Over-Refusal Rate (ORR)**
>
> | Domain     | DeepSeek-v3 | gpt-4o | gpt-5 | Gemini-2.5-flash | o4-mini |
> |:----------|------------:|------:|-----:|------------------:|-------:|
> | civics     | 4.34 | 4.82 | 3.47 | 3.94 | 1.74 |
> | creative   | 6.58 | 2.43 | 4.87 | 3.98 | 2.63 |
> | education  | 7.38 | 6.55 | 7.92 | 6.70 | 8.86 |
> | finance    | 6.78 | 5.02 | 3.52 | 6.16 | 8.14 |
> | general    | 4.99 | 8.30 | 8.39 | 9.06 | 14.97 |
> | health     | 2.58 | 5.73 | 5.53 | 7.04 | 4.65 |
> | security   | 22.01 | 13.96 | 8.21 | 14.28 | 15.10 |
> | technology | 10.72 | 8.92 | 9.83 | 9.73 | 3.21 |
> | **overall**| **6.63** | **6.02** | **6.16** | **6.75** | **6.51** |

---

### Official Review · Reviewer_UihV · 2025-11-01

**Soundness:** 2
**Presentation:** 3
**Contribution:** 2
**Rating:** 4
**Confidence:** 3

**Summary:**

This paper introduces DeepResearchGuard, a safety framework for deep research agents that decomposes complex queries, iteratively searches the literature, and generates structured reports. The authors identify two gaps in the current deep research agent framework: (1) evaluation focuses on QA accuracy rather than report quality, and (2) a lack of stage-specific safeguards allows harmful content to propagate through the multi-stage pipeline.

DeepResearchGuard addresses these by implementing guards at four stages (input, plan, research, output), each with tailored taxonomies, memory-augmented classification, and human-in-the-loop intervention. The authors also contribute DRSafeBench, an 828-query benchmark combining adversarial and benign examples. Experiments across gpt-4o, Gemini-2.5-flash, DeepSeek-v3, and o4-mini show average defense success rate improvements of 18.16% while reducing over-refusal by 6%.

**Strengths:**

1. The four-stage framework design is clear and practical. Each guard has a clear taxonomy, severity-based actions, and memory retrieval for consistency.

2. The author did a comprehensive experimental evaluation. The author tests four baseline LLMs. Then, the author did ablation studies to show that the input guard contributes the most. Also, guard model sensitivity analysis reveals safety-performance tradeoffs

3. In this paper, the author further introduced  DRSafeBench, which has 828 queries and covers diverse failure modes, including synthesized cases for low-quality/format errors.

4. The confidence-based escalation mechanism with user override options balances automation with human judgment.

**Weaknesses:**

1. I have some concerns about the attacking method. Reference poisoning is simulated by rewriting content, but it may be hard to reflect real attacks. Also, no evaluation against actual jailbreaks or adversarial attacks designed to evade the guards.

2. For the report quality evaluation, the author only takes the 1-5 scores for coherence, credibility, etc., and then use LLM-as-judge to evaluate. There is no human evaluation. As safety research work, maybe using human evaluation is important.

3. The paper doesn't compare against LlamaGuard, WildGuard, or other moderation systems that could be adapted for this task.

4. The author didn't evaluate the framework on recent LLMs. Typically, as the model is updated, its safeguards are also upgraded. So, evaluating new model is also necessary for this work. For example, how DeepResearchGuard compares to simply using a stronger base model (e.g., gpt-5 family for research instead of o4-mini + guards)?

**Questions:**

See in Weaknesses.

---

> ### Author Response · Authors · 2025-11-22
> **Response (Part I, W1 and W2)**
>
> **W1**: Thanks for the insightful comment. **Reference poisoning is slightly different from general jailbreak.**
>
> **Jailbreak/Adversarial attack**: Most existing jailbreak and adversarial attack studies focus on the query, because it is the primary interface between the user and the LLM and largely determines whether the model can be manipulated. In these settings, the “jailbreak” behavior is an intentional human action aimed at cheating the LLM.
>
> **Reference**: In contrast, reference collection in a deep research system is a more passive process driven by the LLM through similarity search and retrieval algorithms for web search. Therefore, we can tell that the reference includes some malicious or inappropriate content, but it is tough to determine whether it is an "adversarial" generated reference or a jailbreak, and it also does not fit that the human "intentionally" wants to fool the safeguard system.
>
> The reference threat model is that an attacker mixes harmful content, malicious advertisements, misleading information, or low-credibility sources into the retrieved references. Such content is “malicious” in the sense that it is not accurate or trustworthy enough to support the reasoning chain. To address this, our research guard module explicitly evaluates both the URL and the content of each reference. If a reference is flagged as potentially malicious URL or content, we assign it very low scores on helpfulness, authority, and timeliness, and include this assessment to the reporter. We are not simply excluding potentially malicious references, considering that they may still have some meaningful information, or the LLM evaluation can be wrong. During report generation, the system is encouraged to prioritize high-scoring references and to downweight or ignore those with low scores.
>
> **W2**:
> Thanks for the insightful comments. **We have added the section to compare human evaluation with the LLM-as-judge. Please refer to Appendix G.5**, as described below:
>
> **Section G.5 Reference and Report Quality Evaluation: Human vs. LLM-as-a-Judge**
>
> We recruit 11 CS PhD students as human evaluators for both reference and report quality. We randomly sample 30 references and 30 reports, and ask them to rate (on the same 1–5 Likert scale) reference helpfulness, authority, and timeliness, as well as report coherence, credibility, safety, depth, and breadth. The results are summarized in Tables G.5 and G.6 for reports and references, respectively. For report quality, the LLM-as-judge obtains an average score of 4.48 (±0.42) compared to the human average of 4.04 (±0.45); the gap between LLM and human is modest (e.g., +0.25 for coherence and +0.40–0.56 for the other dimensions). For reference quality, the overall reference score is 4.02 for the LLM and 3.38 for humans, with the largest discrepancy arising in timeliness. The main reason is that some references do not clearly display publication dates, making it difficult for human evaluators to reliably assess timeliness. In contrast, the LLM can leverage its pretrained knowledge and textual cues in the content to infer recency, which supports more confident timeliness judgments. Overall, these results indicate that the LLM-as-a-judge is reasonably aligned with human evaluation, though it is slightly more positive and tends to assign somewhat higher quality scores.
>
> **Report quality: LLM-as-a-judge vs Human**
>
> | Metric      | LLM Mean (STD)    | Human Mean (STD)  | Δ Mean (LLM–Human) |
> |------------|-------------------|--------------------|---------------------|
> | coherence  | 4.73 (± 0.22)     | 4.48 (± 0.39)      | +0.25               |
> | credibility| 3.73 (± 0.41)     | 3.33 (± 0.58)      | +0.40               |
> | safety     | 4.97 (± 0.18)     | 4.56 (± 0.33)      | +0.41               |
> | depth      | 4.52 (± 0.67)     | 3.96 (± 0.89)      | +0.56               |
> | breadth    | 4.43 (± 0.52)     | 3.88 (± 0.65)      | +0.55               |
> | **average**| **4.48 (± 0.42)** | **4.04 (± 0.45)**  | **+0.43**           |
>
> **Reference quality: LLM-as-a-judge vs Human**
>
> | Metric       | LLM Mean (STD)    | Human Mean (STD)  | Δ Mean (LLM–Human) |
> |-------------|-------------------|--------------------|---------------------|
> | Helpfulness | 3.65 (± 1.41)     | 3.07 (± 1.39)      | +0.58               |
> | Authority   | 4.26 (± 0.67)     | 3.65 (± 1.32)      | +0.61               |
> | Timeliness  | 4.15 (± 0.89)     | 3.40 (± 1.21)      | +0.75               |
> | **Overall** | **4.02 (± 0.64)** | **3.38 (± 1.10)**  | **+0.64**           |

---

> ### Author Response · Authors · 2025-11-22
> **Response (Part II, W3)**
>
> **W3**: Thanks for the insightful comment. **We would like to clarify that the LlamaGuard, WildGuard are not designed for deep research work**. The input and output guard of them are isolated and the steps such as planning and collecting references are not protected. Therefore, they are only suitable for LLM but fail to cover the complexity of deep research. We have discussed this issue in the section “LLM, Agent, and Deep Research Safety” in Section 2 (Related Work). In this section, we also cite LlamaGuard and WildGuard. Specifically, we state that:
>
> “Existing safeguards mostly target a single interaction surface: LLM-level moderation screens prompts and responses for harmful content or jailbreaks. Although multi-agent work extends protection to monitor tools, it lacks systematic integration and operates in isolation rather than as a cohesive defense system. In contrast, deep research unfolds over multiple stages (input, planning, research, report) and introduces distinct risks, such as plan drift and web-retrieval threats that bypass single-turn moderation. Hence, the guardrails designed for LLMs and single-turn agents are insufficient for deep research systems.”
>
> We also agree with the reviewer that this claim can include more details, so we would like to refine the last sentence to:
>
> “Hence, the guardrails designed for LLMs and single-turn agents without systematic integration, such as LlamaGuard [1], WildGuard [2], BingoGuard [3], or multi-agent frameworks such as G-Safeguard [4], are insufficient for deep research systems.”
>
> We think this would help the reader can understand the difference between our work and the previous work.
>
> **Reference**
>
> [1] **Llama guard: Llmbased input-output safeguard for human-ai conversations**. https://doi.org/10.48550/arXiv.2312.06674.
>
> [2] **Wildguard: Open one-stop moderation tools for safety risks, jailbreaks, and refusals of llms**. https://papers.nips.cc/paper_files/paper/2024/hash/0f69b4b96a46f284b726fbd70f74fb3b-Abstract-Datasets_and_Benchmarks_Track.html
>
> [3] **Bingoguard: LLM content moderation tools with risk levels**. https://openreview.net/forum?id=HPSAkIHRbb
>
> [4] **G-safeguard: A topology-guided security lens and treatment on llm-based multiagent systems**. https://aclanthology.org/2025.acl-long.359/

---

> ### Author Response · Authors · 2025-11-22
> **Response (Part III, W4)**
>
> **W4**: Thanks for the insightful comment. **To address this, we have added experiments using gpt-5**. As shown in the updated tables, DeepResearchGuard still consistently improves the safety of gpt-5. The DSR increases from 47.34% to 57.36% while the ORR drops from 15.70% to 6.16%. This pattern is obtained as in the previous experiments, demonstrating that DeepResearchGuard can help the baseline model filter harmful content and also decrease the ORR. The more advanced model with better safeguards would prevent harmful content but also lead to a higher ORR. Within the GPT family, we can discover that the gpt-5 already achieves a higher DSR than gpt-4o (47.34% vs. 32.65%), but this comes at the cost of a larger ORR (15.70% vs. 3.98%).
>
> We also include the stage-wise analysis with gpt-5 and observe that the guard maintains high F1 and low FNR across all stages (input, plan, reference, and output). We have incorporated these new results into Tables 2 and 3 in the new version. This result can support that, DeepResearchGuard remains effective even when the underlying research model is already strongly safety-tuned.
>
> | Baseline ± Guard        | DSR (%) ↑ | ORR (%) ↓ |
> |-------------------------|-----------|-----------|
> | gpt-4o                  | 32.65     | 3.98      |
> | +DeepResearchGuard      | 53.98     | 6.02      |
> | Gemini-2.5-flash        | 43.37     | 13.49     |
> | +DeepResearchGuard      | 60.96     | 6.75      |
> | DeepSeek-v3             | 45.66     | 15.18     |
> | +DeepResearchGuard      | 58.07     | 6.63      |
> | o4-mini                 | 32.41     | 13.73     |
> | +DeepResearchGuard      | 53.73     | 6.51      |
> | gpt-5                 | 47.34     | 15.70     |
> | **+DeepResearchGuard**      | **57.36**     | **6.16**      |
>
>
> | Model            | Input F1 (↑) | Input FPR (↓) | Input FNR (↓) | Plan F1 (↑) | Plan FPR (↓) | Plan FNR (↓) | Reference D@1 (↑) | Reference D@All (↑) | Output F1 (↑) | Output FPR (↓) | Output FNR (↓) |
> |------------------|-------------:|--------------:|--------------:|------------:|-------------:|-------------:|-------------------:|--------------------:|--------------:|---------------:|---------------:|
> | gpt-4o           | 0.66         | 0.08          | 0.45          | 0.93        | 0.17         | 0.10         | 0.29               | 0.26                | 0.72          | 0.04           | 0.41           |
> | Gemini-2.5-flash | 0.73         | 0.08          | 0.36          | 0.88        | 0.58         | 0.15         | 0.24               | 0.24                | 0.58          | 0.04           | 0.57           |
> | DeepSeek-v3-0324 | 0.74         | 0.08          | 0.34          | 0.91        | 0.33         | 0.13         | 0.16               | 0.15                | 0.69          | 0.00           | 0.48           |
> | o4-mini          | 0.72         | 0.10          | 0.36          | 0.93        | 0.17         | 0.10         | 0.12               | 0.11                | 0.56          | 0.00           | 0.61           |
> | **gpt-5**         | **0.74**         | **0.08**          | **0.33**          | **0.91**        | **0.16**         | **0.11**         | **0.35**               | **0.33**                | **0.68**          | **0.06**          | **0.45**           |

---

### Author Response · Authors · 2025-12-01
**The Rebuttal Summary and Highlighted Key Issues for AC‘s and SAC's Reference**

Dear Program Chairs, Senior Area Chairs, and Area Chairs,

We have carefully updated the paper to include additional analysis in response to the reviewers' comments. We are confident that our rebuttal has fully addressed the reviewer's comment.

**Before the OpenReview incident, we only received a response from Reviewer `VgF5`**. After our detailed response, the concern and the question from the reviewer have been well-addressed.

**We emphasize that the feedback and comments from all reviewers focused primarily on requests for additional analysis across different dimensions of DeepResearchGuard and the DRSafeBench benchmark**, and no reviewers commented on any factual or fundamental errors in our framework. We have confidence that our work has contributed to improved safety in the deep research workflow, and we have added the following summary to highlight the new revision improvement and how the reviewer's concerns and questions are addressed.

We want to highlight that, **regarding the Reviewer `zyxB` and `VgF5` 's comments, we have a strong suspicion that the reviews were largely AI-generated, with or without very minimal human editing**. We will submit a confidential message to the PC, SAC, and AC regarding this issue with the evidence we collected to support our suspicion.

Below is a summary of the  rebuttal and revision ( In the PDF, we highlight the newly added content in blue):

### **Reviewer `UihV`** ###

**W1**: We clarified that the reference poisoning does not have a subjective or intentionally conceived like a human "jailbreak". Therefore, we can tell the reference is malicious but hard to describe as a jailbreak example.

**W2**: We conducted a study with 11 PhD students to rate references and reports. The performance of humans and LLMs as judges is consistent, but LLMs tend to be more positive than humans  (e.g., Human avg 4.04 vs. LLM avg 4.48).

**Revision based on W2**: We added Appendix G.5 (Reference and Report Quality Evaluation: Human vs. LLM-as-a-Judge).

**W3**: We explained that baselines like LlamaGuard are designed for single-turn and fail to monitor the intermediate "Plan" and "Reference" stages of the deep research workflow. We argue that without stage-specific integration, these models cannot prevent error propagation in complex workflows.

**Revision based on W3**: We revised Section 2 to specifically describe that LlamaGuard and WildGuard are not suitable for protecting deep research workflows.

**W4**: We tested the framework on GPT-5 and found that DeepResearchGuard increased the DSR from 47.34% to 57.36% while lowering the ORR. This proves that even advanced models benefit from stage-specific architectural guardrails.

**Revision based on W4**: We updated Tables 2 and 3 to include the GPT-5 performance results.


### **Reviewer `FuFk`** ###

**W1 & Q1**: We clarified that we did not synthesize data from scratch but used datas from previous benchmark and only used LLMs to rewrite benign prompts into specific failure format while keeping the original meaning.

**Revision based on W1 & Q1**: We added Appendix E.4, which representative data samples to demonstrate data quality.

**W2**: We acknowledged additional times of latency per query but justified this as a necessary trade-off for high-stakes research tasks where safety is very important. We found the cost is linear regarding the number of stages and that early detection saves compute by stopping harmful queries early.

**Revision based on W2**: We revised Appendix G.2 to provide a detailed runtime and latency analysis.

**W3 & Q3**: We explained that the four-stage design mirrors human research workflows (reviewing plans, checking sources) and is the minimum necessary complexity to catch risks that bypass single-step guards. We noted that a simpler design would fail to maintain the cross-stage safeguard mechanism against the harmful content.

**Revision based on W3 & Q3**: We added Appendix A.6 to describe the design logic and necessity of the architecture for DeepResearchGuard.

**W4**: We provided a formal definition of "Deep Research" in our work. This creates a clear scope for where our framework is applicable.

**Revision based on W4**: We added a definition section to Appendix B.1.

**Q2**: We present domain-wise performance and find consistent protection across 8 domains.

**Revision based on Q2**: We added Appendix E.5 and Appendix G.7 for the DRSafeBench distribution, and the domain-wise DSR and ORR analysis, respectively.

We appreciate all reviewers’ time and insightful feedback, which helped us improve the work. We also sincerely appreciate the AC's, SAC's, and PC's support and efforts in ensuring a fair and rigorous reviewing process.

We hope the current revisions address all of the concerns. We are happy to provide any further clarification.

Best regards,

DeepResearchGuard Author Team

---

> ### Author Response · Authors · 2025-12-01
> **The Rebuttal Summary and Highlighted Key Issues for AC‘s and SAC's Reference - Part II**
>
> ### **Reviewer `VgF5`** ###
>
> **W1**: Our ablation study showed that adding guards increases report quality scores by filtering out noise.
>
> **W2**: We initially had analyzed reference scoring across models in the first version and found no bias in how different models rate "Helpfulness," "Authority," or "Timeliness."
>
> **Revision based on W2**: We added the description of the calculation principle in Appendix G.1.
>
> **W3**: We measured the human intervention rate and found that human intervention is only required for 5.4%–11.4% of cases, mainly during the research stage.
>
> **Revision based on W3**: We added Appendix G.4 to report human intervention rates and time usage.
>
> **W4**: We provided a breakdown of the open-domain tasks to show the diversity of the DRSafeBench. We confirmed the the DRSafeBench covers varied categories to ensure broad evaluation.
>
> **Revision based on W4**: We added open-domain coverage statistics in Appendix E.5.
>
> **Q2**: We explained that confidence thresholds (0.5, 0.7, 0.8) were selected based on the confidence distribution percentiles from the experiment to maintain a human intervention rate of ~10%.
>
> **Revision based on Q2**: We added Appendix F.3, which describes the methodology for threshold selection.
>
> **Q4**: We provided a granular breakdown of token usage and claim that refusal types per stage have been reported in the original version.
>
> **Revision based on Q4**: We added Appendix G.6 for the token usage analysis.
>
>
> ### **Reviewer `zyxB`** ###
>
> **W1 & W2**: The reviewer criticized the lack of statistical grounding for "Reasoning Depth" and "Factuality" claims (e.g., "+15% factuality"). We clarified that **we do not use these metrics** nor make those specific numerical claims in the paper.
>
> **W3**: We refuted the claim that our work lacks novelty. We explained that prior works focus on single-turn interactions, whereas DeepResearchGuard is the first to address the specific "Plan" and "Reference" risks inherent to multi-stage deep research, which generic LLM oversight misses.
>
> **W4**: The reviewer asked for proof of "risk subadditivity." We clarified that this is an empirical system paper, not a theoretical proof, and that "risk subadditivity" is an economics term not standard in this domain.
>
> **W5**: The reviewer criticized a lack of details on "RLHF" and "Model Size." We clarified that our method is prompt-based, involving **no model training or RLHF**, and therefore model size/training details are not applicable.
>
> **W6**: The reviewer claimed the benchmark was self-referential and biased. We corrected this by explaining that DRSafeBench is curated from **external, top-tier benchmarks** and partially generated by us.
>
> **W7**: The reviewer claimed the cost increases superlinearly. We corrected this to show the cost is **linear** regarding the fixed number of stages. We also explained that because each stage uses different criteria (Plan vs. Reference), the composite guard does not become a "biased projection" of a single prior.
>
> **W8**: The reviewer said that the ethical discussion was very generic. We countered that **Appendix A.1** already provides specific descriptions for ethical issues.
>
> **Q1**: The reviewer asked about conflicts involving a "Process Guard." We clarified that **no such component exists** in our framework (we use Plan/Research guards).
>
> **Q2**: The reviewer asked for evidence that "intermediate guard checks" prevent rather than delay unsafe outcomes. We clarified that **no component named "intermediate guard" exists** in our system. We prove that the guard removes the harmful content from the F1 score.
>
> **Q3**: The reviewer asked if the benchmark would be public. We confirmed that DRSafeBench will be publicly released. We also directed them to the representative data samples we added to the paper.
>
> **Q4**: The reviewer asked if guards degrade reasoning. We mentioned that Figure 3, which shows that report quality (depth, coherence) actually *increases* with guards enabled, proving that filtering noise helps the reasoning and not degrades it.
>
> **Q5**: The reviewer asked about "automatic retrieval verification." We clarified that we do not use this term in our work.

---

### Note · Authors · 2026-01-06

**Comment:**

Dear Program Chairs,
We would like to withdraw our submission from ICLR 2026. Thank you for your time and consideration.

**Withdrawal Confirmation:**

I have read and agree with the venue's withdrawal policy on behalf of myself and my co-authors.